# Prognostic significance of frequent CLDN18-ARHGAP26/6 fusion in gastric signet-ring cell cancer

Yang Shu[1,2], Weihan Zhang[1], Qianqian Hou[2], Linyong Zhao[1], Shouyue Zhang[2], Jiankang Zhou[3], Xiaohai Song[1], Yan Zhang[4], Dan Jiang[5], Xinzu Chen[1], Peiqi Wang[6], Xuyang Xia[2], Fei Liao[2], Dandan Yin[2], Xiaolong Chen[1], Xueyan Zhou[2], Duyu Zhang[2], Senlin Yin[3], Kun Yang[1], Jianping Liu[5], Leilei Fu[3], Lan Zhang[3], Yuelan Wang[2], Junlong Zhang[7], Yunfei An[7], Hua Cheng[8], Bin Zheng[8], Hongye Sun[8], Yinglan Zhao[3], Yongsheng Wang[4], Dan Xie[2,3], Liang Ouyang[3], Ping Wang[9], Wei Zhang[10], Meng Qiu[11], Xianghui Fu[3], Lunzhi Dai[3], Gu He [ID] [3], Hanshuo Yang[3], Wei Cheng[3], Li Yang[3], Bo Liu[3], Weimin Li[12], Biao Dong[3], Zongguang Zhou[1], Yuquan Wei[3], Yong Peng[3], Heng Xu [ID] [2,3,7] & Jiankun Hu[1]

Signet-ring cell carcinoma (SRCC) has specific epidemiology and oncogenesis in gastric cancer, however, with no systematical investigation for prognostic genomic features. Here we report a systematic investigation conducted in 1868 Chinese gastric cancer patients indicating that signet-ring cells content was related to multiple clinical characteristics and treatment outcomes. We thus perform whole-genome sequencing on 32 pairs of SRC samples, and identify frequent CLDN18-ARHGAP26/6 fusion (25%). With 797 additional patients for validation, prevalence of CLDN18-ARHGAP26/6 fusion is noticed to be associated with signet-ring cell content, age at diagnosis, female/male ratio, and TNM stage. Importantly, patients with CLDN18-ARHGAP26/6 fusion have worse survival outcomes, and get no benefit from oxaliplatin/fluoropyrimidines-based chemotherapy, which is consistent with the fact of chemo-drug resistance acquired in CLDN18-ARHGAP26 introduced cell lines. Overall, this study provides insights into the clinical and genomic features of SRCC, and highlights the importance of frequent CLDN18-ARHGAP26/6 fusions in chemotherapy response for SRCC.

[1] Department of Gastrointestinal Surgery, Institute of Gastric Cancer, State Key Laboratory of Biotherapy, West China Hospital, Sichuan University and Collaborative Innovation Center, 610041, Chengdu, Sichuan, China. [2] Precision Medicine Center, State Key Laboratory of Biotherapy and Precision Medicine, Key Laboratory of Sichuan Province, West China Hospital, Sichuan University and Collaborative Innovation Center, 610041 Chengdu, Sichuan, China. [3] State Key Laboratory of Biotherapy, West China Hospital, Sichuan University and Collaborative Innovation Center, 610041 Chengdu, Sichuan, China. [4] Department of Thoracic Oncology, Cancer Center, State Key Laboratory of Biotherapy, West China Hospital, Sichuan University, 610041 Chengdu, Sichuan, China. [5] Department of Pathology, West China Hospital, Sichuan University, 610041 Chengdu, Sichuan, China. [6] State Key Laboratory of Oral Diseases, West China Hospital of Stomatology, Sichuan University, 610041 Chengdu, China. [7] Department of Laboratory Medicine/Research Center of Clinical Laboratory Medicine, West China Hospital, Sichuan University, 610041 Chengdu, Sichuan, China. [8] WuxiNextCODE, 200131 Shanghai, China. [9] Department of Neurology, Albert Einstein College of Medicine, Bronx, NY 10461, USA. [10] Department of Clinical Pharmacology, Hunan Key Laboratory of Pharmacogenetics, Xiangya Hospital, Central South University, 410008 Changsha, China. [11] Department of Abdominal Oncology, Cancer Center, West China Hospital, Sichuan University, 610041 Chengdu, Sichuan, China. [12] Department of Respiratory and Critical Care Medicine, West China Hospital, Sichuan University, 610041 Chengdu, Sichuan, China. These authors contributed equally: Yang Shu, Weihan Zhang, Qianqian Hou, Linyong Zhao. These authors jointly supervised this work: Yong Peng, Heng Xu, Jiankun Hu. Correspondence and requests for materials should be addressed to Y.P. (email: pengyong10@hotmail.com) or to H.X. (email: xuheng81916@scu.edu.cn) or to J.H. (email: hujkwch@126.com)

Gastric cancer is one of the most common cancers and leading causes of cancer-related mortality in the world, particularly in China[1,2]. Multiple subtypes are classified, such as intestinal and diffuse types according to Lauren's classification[3–5], and diffuse type has significantly worse treatment outcomes than intestinal type[6]. To figure out the molecular mechanisms for tumorigenesis and heterogeneity of gastric cancer at the molecular level, large efforts have been taken to characterize the comprehensive genomic features through high-throughput genomic screening[3,7–14], and multiple driver alterations have been identified. These altered genes are either commonly identified in other cancers (e.g., *TP53*, *PIK3CA*, *CDH1*, *SMAD4*) or relatively specific in gastric cancer (e.g., *RHOA*)[3,7,15]. As part of The Cancer Genome Atlas (TCGA) project, four subtypes (i.e., EBV, MSI, GS, and CIN) have been systematically analyzed separately in the largest gastric patient cohort ($N = 295$) including subtypes of Epstein–Barr virus infected (EBV), microsatellite instability (MSI), genomically stable (GS), and chromosomal instability (CIN). It is found that the frequency of *PIK3CA* mutations is high (80%) in EBV subtype but low in CIN subtype (3%), while *ARID1A*, *RHOA*, and *CDH1* mutations are prevalent in GS subtype[3], which has been validated in diffuse type of gastric cancer[7]. Additionally, recurrent structure rearrangement has been observed between *CLDN18* and *ARHGAPs* (i.e., *ARHGAP26* or *ARHGAP6*)[3,16], which is also enriched in diffuse type, and mutually exclusive with *RHOA* mutations[3]. Despite of significant ethnic differences of gastric cancer in terms of prevalent and treatment outcomes[17], no significant difference for the frequent mutated genes has been identified on the basis of ethnic origin in TCGA study[3]. Additionally, no systematical investigation on the association of genetic alterations with clinical features has been done due to the lack of long-term follow-up information for TCGA gastric cancer cohort.

Besides Lauren's classification, gastric cancer with at least 50% of signet-ring cell in the pathologic specimen is defined as signet-ring cell carcinoma (SRCC) based on the microscopic characteristics according to World Health Organization (WHO) classification[18–20]. Although all SRCCs belong to, and account for less than half of diffuse type[5], distinct epidemiology and oncogenesis of SRCC have been observed including female/male ratio, tumor location, tumor stage, etc.[19,21] SRCC is positively related to survival outcomes in early gastric cancer[22], however, paradoxically associated with worse prognosis compared to non-SRCC in advanced tumor stage[18,19], and may have different chemosensitivity profiles[19,23–25]. Although a few of the SRCC patients may be analyzed as diffuse type in previous studies[3], no systematical study has been done to investigate the comprehensive molecular characterizations of SRCC due to the heterogeneity and low content of signet-ring cells in most tumor samples.

In this study, we systematically investigate the specific clinical features of SRCC, and characterize the genomic features of SRCC tumors with >80% presence of signet-ring cells (defined as HSRCC) through whole-genome sequencing (WGS), to determine clinically relevant (e.g., survival outcomes) frequent genomic alterations in a large patient cohort.

## Results

### Clinical characteristics and prognostic value of SRCC.
In this study, a total of 1868 primary gastric cancer patients who had underwent gastrectomy from 2006 to 2012 were included for analyses (Supplementary Fig. 1 and Supplementary Table 1). SRCC patients were defined according to WHO classification (containing >50% of signet-ring cells in pathologic tumor specimen, $N = 375$ [20.1%]). Further, to investigate the influence of low frequency of signet-ring cells, we divided the rest of the patients into two groups: con-SRCC (containing <50% of signet-ring cells, $N = 556$ [29.8%]) and non-SRCC (no signet ring cell at all, $N = 937$ [50.2%]) (Supplementary Table 2). Consistent with previous reports, we found significant differences between SRCC and non-SRCC patients in terms of multiple clinical characteristics. Not surprisingly, feature values of con-SRCC patients rank between those of SRCC and non-SRCC, indicating the positive association of signet cells content with younger age, higher female/male ratio, advanced tumor stage, lower tumor locations, higher risk of invasion, and higher frequency of diffuse subtypes (Supplementary Table 3).

Survival outcomes were compared among non-SRCC, con-SRCC, and SRCC groups (Fig. 1a) in 1703 out of 1868 patients (91.2%) with fully postoperative follow-up information, Signet-ring cell content was related to shorter survival time in patients with advanced stage but not stage I (Fig. 1b, and Supplementary Fig. 2). Univariate and multivariate survival analyses were conducted, identifying signet-ring cell content as an independent prognostic factor for survival outcomes in gastric cancer, as well as TNM stage, capillary invasion, etc. (Table 1).

### Chemotherapy treatment outcomes of SRCC.
We next investigated the survival outcomes by separating patients into two groups in terms of chemotherapy usage. Not surprisingly, the overall survival rate increased significantly in patients with chemotherapy treatment ($P = 0.002$, Log-rank test, Supplementary Fig. 3a). Considering the signet-ring cell content, non-SRCC and con-SRCC but not SRCC patients got benefit from chemotherapy in all stages (Fig. 1c and Supplementary Fig. 3) or advanced stage only (Fig. 1d), while patients with diffuse type, which all SRCCs belong to, also have significantly longer survival time with chemotherapy introduction (Supplementary Fig. 4), therefore SRCC is considered as an independent prognostic factor for chemotherapy treatment.

### Genomic alterations of SRCC identified by WGS.
Next we sought to identify the genomic features of SRCC due to its distinct clinical characteristics. In this case, we focused on HSRCC, which has >80% presence of signet-ring cells in the tumor specimen as described above (Supplementary Fig. 5), and found that HSRCCs tend to have even worse survival outcomes compared to non-SRCC, with 38% and 73% of 3-year overall survival rate, respectively ($P = 0.001$, Log-rank test, Supplementary Fig. 6). All HSRCC tissue samples for WGS were got from patients enrolled in 2012, and similar association of SRCC with clinical characteristics were observed (Supplementary Table 3). WGS was performed on all available 32 tumor/normal pairs of HSRCC samples, with a mean depth of 66.4×(range from 57.2 to 89.3×) and 40.4× (range from 30.6 to 52.1×) covering 98.3% and 95.3% of reference genome with ≥20 depths in tumors and matched control, respectively. Totally, we identified more than 1000 potentially functional somatic SNVs and 16 INDELs (931 missense, 63 nonsense, 27 splice sites, 6 inframe INDELs, 10 frameshift INDELs). No obvious hypermutant or MSI tumor sample was identified according to sequencing-based MSI determination. Alternatively, patients can be classified into hypomutant and non-hypomutant in terms of total mutations (Fig. 2).

A total of 949 genes have at least one somatic non-silent SNV or small INDEL in coding region (Supplementary Data 1), of which six significantly mutated genes (SMG) were identified, including *TP53* (25%), *CDH1* (15.6%), *PIK3CA* (12.5%), *ERBB2* (6.3%), *LCE1F* (6.3%), and *OR8J1* (6.3%) (Fig. 2 and Supplementary Data 2), but not the well-reported SMGs enriched in

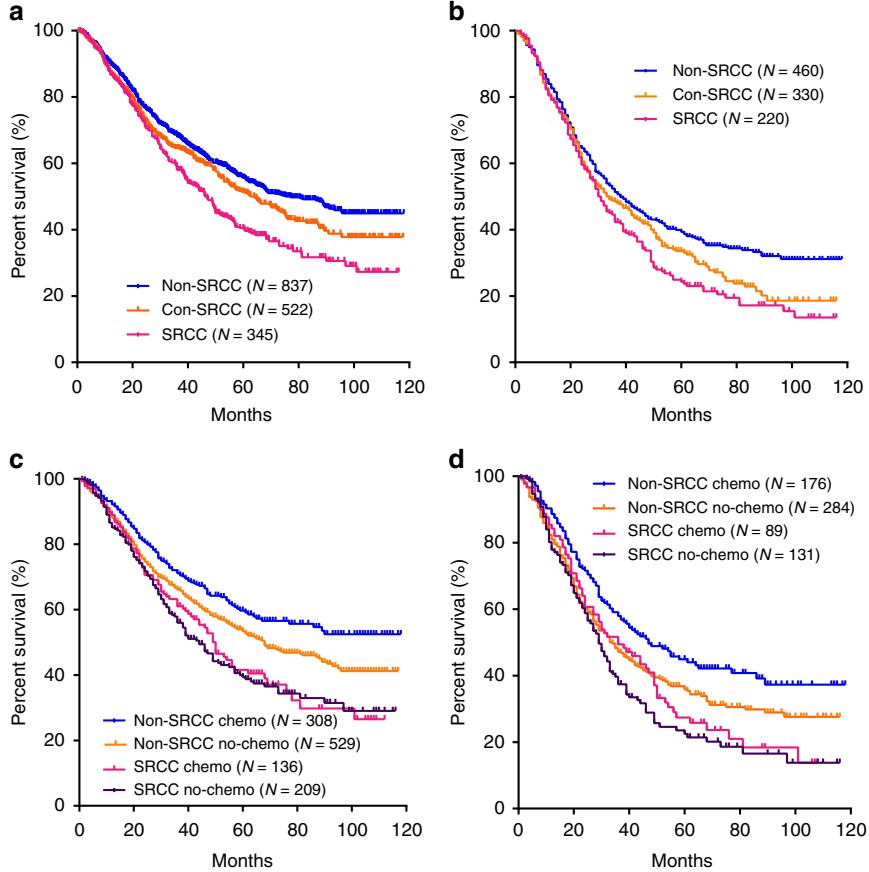

**Fig. 1** Survival outcomes in gastric cancer patients with different signet-ring cell frequency (2006–2012). Survival curves of patients among the non-SRCC group ($N = 837$), con-SRCC group ($N = 522$), and SRCC group ($N = 345$) were illustrated in all patients (**a**), and stages III/IV (**b**). Impact of chemotherapy introduction on survival was also illustrated separately in terms of SRCC content in all patients (**c**), and stage III/IV (**d**)

diffuse type, such as *ARID1A*, *RHOA*, and *SMAD4* (Fig. 2, Supplementary Fig. 7, and Supplementary Data 2), indicating possible distinct genomic features of SRCC from other diffuse type of gastric cancer. Interestingly, despite of low mutation rate in *RHOA*, multiple mutations were identified in its regulatory factors, such as RhoGAPs (GTPase-activating protein, including *ARHGAP1*, *ARHGAP5*, and *ARHGAP26*) or RhoGEFs (GDP/GTP-exchange factor, including *ARHGEF2*, *ARHGEF5*, *ARHGEF33*, and *ARHGEF40*) (Fig. 2 and Supplementary Data 1). With pathway analyses, we identified frequent alterations across multiple pathways in non-hypomutant group (e.g., cell adhesion, enriched score = 8.5, FDR = $8.1 \times 10^{-9}$, Supplementary Data 3), but not hypomutant group (Supplementary Data 4). Further Protein–protein interaction (PPI) network analyses identified 107 additional cell adhesion-related mutant genes (Supplementary Fig. 8), indicating the important role of cell adhesion pathway on SRCC tumorigenesis.

Consistent with the findings of stable genomic characterizations for diffuse type, only 4 out of 32 tumor samples had obvious extensive somatic copy number alterations (SCNAs). Besides the recurrent large-scale copy number gain and loss (Supplementary Fig. 9), 15 out of 32 tumors had foci or extensive of loss at a tumor suppressor (i.e., 46.9% of *FHIT*) (Supplementary Fig. 10A), while recurrent amplifications were observed at 8q24.21 (21.8%, including *MYC*), 10q26.13 (12.5%, including *FGFR2*), 11p13 (12.5%, including *CD44*), 19q12 (9.4%, including *CCNE1*), and 20q13.2 (15.6% including *BCAS1*) (Supplementary Fig. 10B-F), among which *MYC*, *FGFR2*, *CD44*, and *CCNE1* were well-known oncogenes.

We further investigated the somatic SVs (Supplementary Data 5), and identified high frequency of *CLDN18-ARHGAP* fusion (Figs. 2 and 3a), which linked exon5 or downstream of *CLDN18* to exon 12 ($N = 6$) or exon 10 ($N = 1$) of *ARHGAP26*, or to exon 2 ($N = 1$) of *ARHGAP6* (Fig. 3b). With Sanger sequencing validation in cDNA level, we noticed that the *ARHGAPs* splicing acceptor activated a cryptic splicing site before the stop codon of *CLDN18* in exon5 (Fig. 3b), and validated the result in another 65 out of 797 patients (Supplementary Data 6). With activation of this cryptic splicing site, truncated CLDN18 (lost last 11 amino acids) and ARHGAP26/6 (lost the first to the translocated exons) are predicted to be inframely fused. Moreover, patients with *CLDN18-ARHGAP26/6* fusion tended to have *ARHGAPs* or *ARHGEFs* mutations ($P = 0.04$), but were mutually exclusive with *CDH* gene (i.e., *CDH1*, *CDH4*, *CDH6*, and *CDH8*) mutations (Fig. 2).

Finally, we integrated all genes involved in SNV/INDEL, SVs, and SCNA (focal alterations containing no more than three genes), and performed cluster and pathway analyses, cell adhesion category still ranks the top (Supplementary Data 7).

**Correlation of genomic features to clinical characteristics.** Logistic regression model was used to evaluate the relationship between clinical characteristics and genetic alterations. Mutation rate was positively related to age at diagnosis ($P = 0.04$, Logistic regression test), with an average age of 49 ($\pm 13$ years) and 57 ($\pm 13$ years) in hypomutant and non-hypomutant group, respectively. For highly mutated genes, *PIK3CA* mutations were

**Table 1 Independent factors for survival prediction multivariate analysis of patients**

| Variables | Characteristics | Univariate HR (95% CI) | P value | Multivariate HR (95% CI) | P value |
|---|---|---|---|---|---|
| SRCC status | Non-SRCC vs. Con-SRCC | 1.16 (0.99–1.36) | 0.06 | 1.16 (0.99–1.36) | 0.06 |
| | Non-SRCC vs. SRCC | 1.48 (1.25–1.75) | <0.001 | 1.45 (1.22 –1.71) | <0.001 |
| Age (years) | <60 vs. ≥60 | 1.17 (1.02–1.34) | 0.021 | | |
| Gender | Male vs. Female | 0.98 (0.84–1.13) | 0.74 | | |
| Tumor size (cm) | <5 vs. ≥5 | 2.62 (2.26–3.03) | <0.001 | 1.49 (1.27–1.75) | <0.001 |
| Tumor location | Non-AEG vs. AEG | 1.21 (1.04–1.40) | 0.012 | | |
| Tumor grade | G1–2 vs. G3–4 | 1.61 (1.34–1.94) | <0.001 | | |
| Residual degree | R0 vs. R1/R2 | 2.98 (2.52–3.54) | <0.001 | 1.41 (1.17–1.71) | <0.001 |
| T stage | T1–3 vs. T4 | 3.27 (2.77–3.86) | <0.001 | | |
| N stage | N0 vs. N1–3 | 3.43 (2.84–4.14) | <0.001 | | |
| M stage | M0 vs. M1 | 3.67 (3.09–4.36) | <0.001 | | |
| TNM stage | I vs. II | 2.00 (1.48–2.70) | <0.001 | 1.88 (1.38–2.55) | <0.001 |
| | I vs. III | 5.08 (3.93–6.55) | <0.001 | 3.88 (2.96–5.10) | <0.001 |
| | I vs. IV | 11.64 (8.74–15.51) | <0.001 | 7.58 (5.50 –10.42) | <0.001 |
| Nervous invasion | Negative vs. Positive | 1.62 (1.30–2.02) | <0.001 | | |
| Capillary invasion | Negative vs. Positive | 1.62 (1.39–1.88) | <0.001 | 1.20 (1.03–1.41) | 0.019 |
| Extranodal metastasis | Negative vs. Positive | 2.50 (2.11–2.95) | <0.001 | 1.28 (1.07–1.52) | 0.007 |
| Chemotherapy | With vs. Without | 1.25 (1.09–1.43) | 0.002 | 1.46 (1.27–1.68) | <0.001 |

P value and ORs were estimated by the Cox regression model

HR Hazard ratio, 95% CI 95% confidence interval of the risk ratio, non-SRCC cancers without signet-ring cells, con-SRCC cancers with <50% presence of signet-ring cells, SRCC cancers with >50% presence of signet-ring cells, U upper, M middle, L lower, AEG adenocarcinomas of the esophagogastric junction

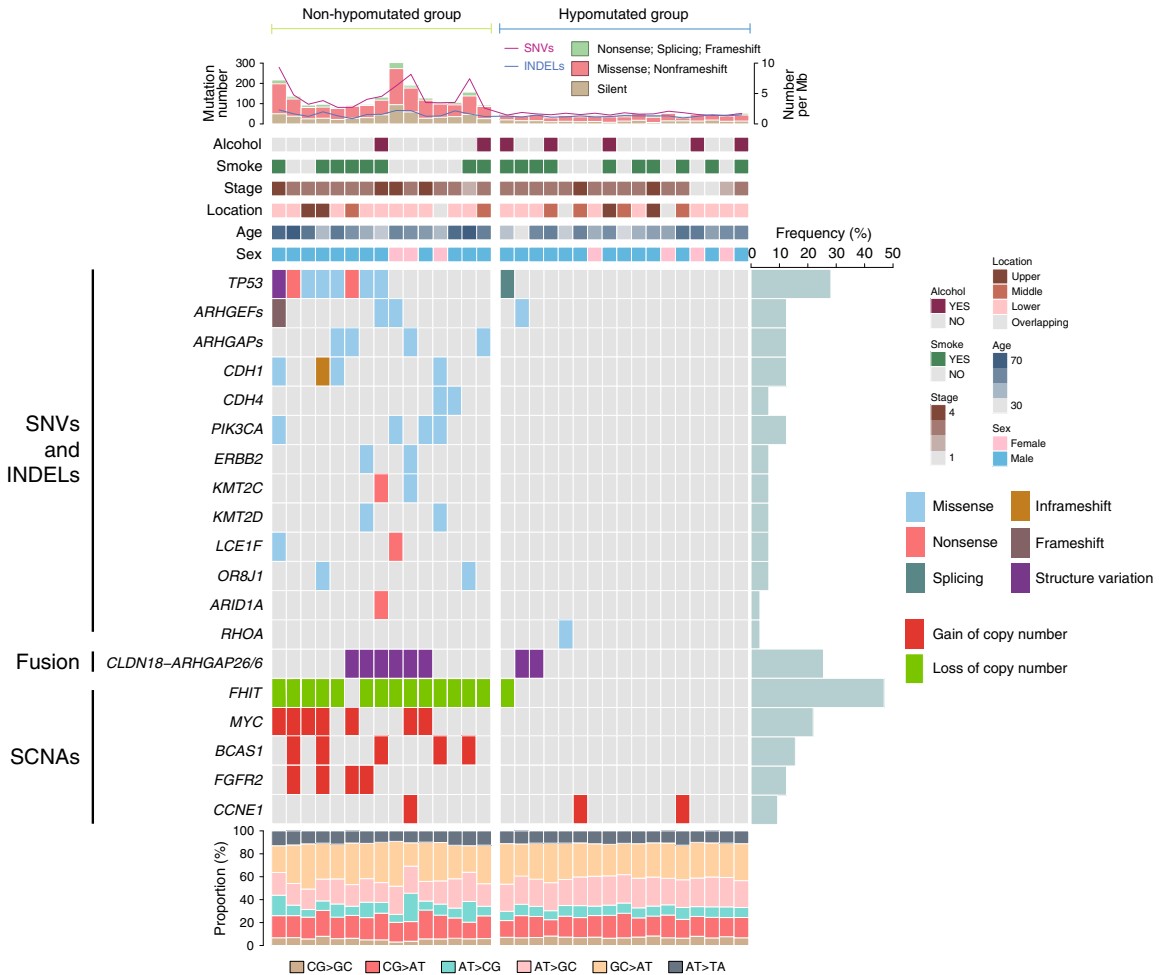

**Fig. 2** Landscape of key genetic alterations in HSRCC of gastric cancer. The patient samples are shown on the x-axis. Information of mutation rate, alcohol history, smoke history, tumor location, stage, patient age, and sex are shown on the top of y-axis, followed by the key genetic alterations including significant mutated genes. Frequency of each alteration was illustrated on the right of the mutation heat plot

significantly enriched in patients in M1 stage ($P = 0.001$, Logistic regression test) and nervous invasion ($P = 0.002$, Logistic regression test). *TP53* mutations occurred more frequently in tumors located at upper region of stomach ($P = 0.05$, Logistic regression test).

More importantly, as for the gastric-specific genetic alteration, we noticed that most of *CLDN18-ARHGAP26/6* fusions resulted in *CLDN18/exon5-ARHGAP26/exon12* (58/73), while only a few cases were *CLDN18/exon5-ARHGAP26/exon10* (N = 7/73), *CLDN18/*

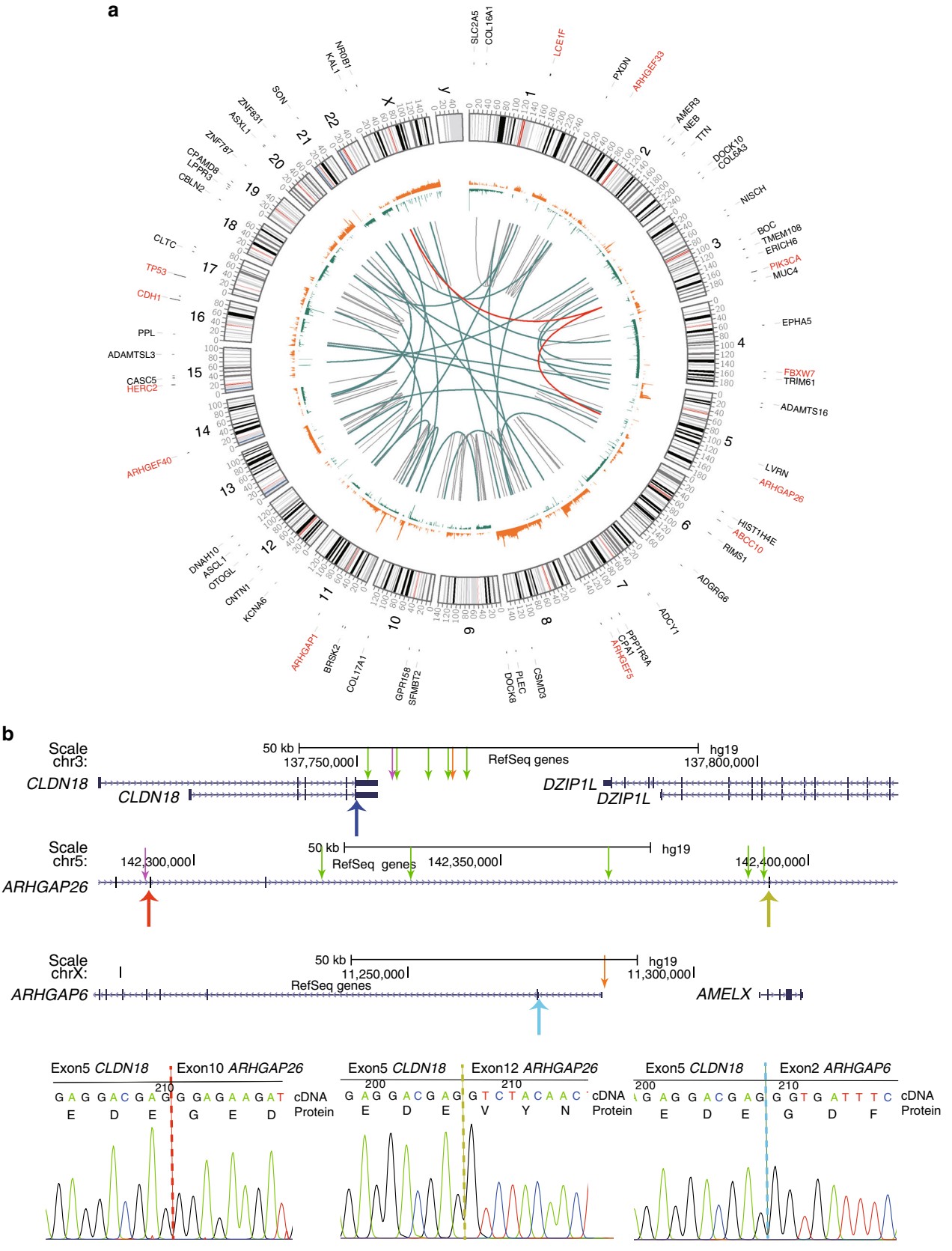

exon4-*ARHGAP26*/exon11 ($N = 1/73$), and *CLDN18*/exon5-*ARH-GAP6*/exon2 ($N = 7/73$). Frequency of such fusion was in parallel with signet-ring cell content in either all patients (2.2% non-SRCC, 11.0% con-SRCC, and 17.0% SRCC, $P = 4.1 \times 10^{-9}$, Logistic regression test) or patients with diffuse type alone (0% non-SRCC, 14.8% con-SRCC, and 19.4% SRCC, $P = 0.003$, Logistic regression test), and positively related to younger age at diagnosis ($51.3 \pm 12.4$ years vs. $60.7 \pm 12.2$ years, $P = 4.2 \times 10^{-10}$, Logistic regression test), female/male ratio (4.6% in male vs. 18.5% in female, $P = 1.7 \times 10^{-9}$, Logistic regression test), advanced TNM stage (1.6% in stage I/II, 9.7% in stage III, and 16% in stage IV, $P = 1.7 \times 10^{-5}$, Logistic regression test), and Lauren's subtypes (1.2% in intestinal, 15.4% in diffuse, $P = 0.005$, Logistic regression test) (Fig. 4a–c and Supplementary Table 4). Interestingly, patients with *CLDN18-ARHGAP26/6* fusions tended to have a higher N stage ($P = 2.2 \times 10^{-7}$, Logistic regression test) and M stage ($P = 0.003$, Logistic regression test), but not T stage ($P = 0.13$, Logistic regression test), indicating this fusion may contribute to tumor metastasis rather than invasion. Despite of the differences of clinical characteristics in TCGA cohort, recurrent *CLDN18-ARHGAP* fusion (13/295) was also significantly enriched in patients at a younger age ($60.2 \pm 10.7$ years vs. $66.8 \pm 10.7$ years, $P = 0.03$, Logistic regression test), females (2.3% in male vs. 8% in female, $P = 0.03$, Logistic regression test), and diffuse type of gastric cancer (1.1% in intestinal, 11.8% in diffuse, $P = 0.001$, Logistic regression test) (Fig. 4a–c and Supplementary Table 4). No significant difference was observed of TNM stage in TCGA patients, possibly because of their older age and ethnic diversity. Since SRCC status are also associated with gender, age and stage, we conducted multivariate analyses and found that the association of *CLDN18-ARHGAP* fusion with these clinical characteristics can only partially explained by SRCC status (Supplementary Table 4). For instance, the frequency of *CLDN18-ARHGAP* fusion raised to 12.1% and 24.4% in male and female patients with SRCC, respectively.

We next evaluated the prognostic potential of the genetic alterations in gastric cancer. Patients with *CLDN18-ARHGAP26/6* fusion had worse survival outcomes compared to fusion-free patients ($P = 0.03$, Cox's regression model, Supplementary Fig. 11A), which can be partially explained by TNM stages (e.g., $P = 0.35$ for patients at stage III and IV, Supplementary Fig. 11B), indicating *CLDN18-ARHGAP26/6* fusion may not be an independent predictor for survival outcomes. To exclude the impact of therapy regimen, we compared patients with and without any chemotherapy treatment. Patients with *CLDN18-ARHGAP26/6* fusion got no benefit from chemotherapy ($P = 0.92$, HR = 1.03, 95% CI: 0.55–1.94, Cox's regression model) compared to fusion-free patients ($P = 0.001$, HR = 1.41, 95% CI: 1.15–1.75, Cox's regression model) after adjusting for stages and SRCC status (Table 2 and Fig. 5), suggesting the potential prognostic value of *CLDN18-ARHGAP26/6* fusion for chemotherapy introduction independent of TNM stages.

**Function of the common fusion in cell lines**. To figure out the role of *CLDN18-ARHGAP26* fusion in gastric cells, we stably expressed *CLDN18-ARHGAP26* in a gastric cancer cell line

(i.e., BGC-823). Compared to control, cells with *CLDN18-ARHGAP26* overexpression got no advantage in cell proliferation (Fig. 6a), but had significantly increased ability of cell migration (Fig. 6b), which is considered as a late event in cancer progression. Moreover, we proceeded to drug response assay by treating the cells with either 5-fluorouracil or oxaliplatin. *CLDN18-ARHGAP26* overexpressed cells exhibited around three folds of resistance to oxaliplatin compared to control ($IC_{50} = 4.6$ [95% CI: 3.2–6.7] vs. 1.6 [95% CI: 0.9–2.7]), and 5-fluorouracil ($IC_{50} = 0.58$ [95% CI: 0.37–0.91] vs. 0.21 [95% CI: 0.13–0.34]) (Fig. 6c), providing further evidence for fusion-induced chemotherapy resistance. Next, *CLDN18-ARHGAP26* was introduced into two additional cell lines (i.e., AGS and MKN-74), and little change has been observed in terms of growth rate and migration ability (Supplementary Fig. 12a and 12b). However, cells with fusion overexpression also exhibited drug resistant to 5-fluorouracil/oxaliplatin in MKN-74 but not AGS cell lines (Supplementary Fig. 12c and 12d), indicating the varied effects of *CLDN18-ARHGAP26* among gastric cells with different genomic backgrounds.

## Discussion

Gastric cancer is one of the most common malignant digestive cancers, and its treatment strategy has been well developed during the past decades[1,2]. SRCC has been firstly noticed for its specific microscopic characteristics, and defined as a high-grade malignancy subtype. In this study, we divided the patients into three groups, and found that the signet-ring cell content was in parallel with higher female/male ratio, younger age, higher risk of serosa invasion, and lymph nodes metastasis, which is consistent with previous reports[18,21]. Moreover, SRCC status is associated with survival outcomes independently in advanced stage only, providing additional evidence to settle the arguments on its prognostic value[21,26–28].

Considering the different chemosensitive profiles of SRCC and non-SRCC in Caucasians[25,29], we firstly systematically investigated the influence of SRCC status on survival outcomes of chemotherapy treatment in Chinese population, and noticed that current chemotherapy strategy can significantly benefit non-SRCC and con-SRCC but the curative effect remains unclear to SRCC patients. Since significant improvement of treatment outcomes in terms of survival rate in patients with diffused types, which contains all SRCC (Supplementary Fig. 4), we considered that determination of SRCC status rather than Lauren's classification may guide the chemotherapy usage in gastric cancer treatment. However, more independent validations in large sample sized are needed, and prospective trials (e.g., ClinicalTrials.gov, NCT01717924)[30] are still expected.

The distinct clinical characteristics and treatment outcomes of SRCC indicate the importance of systematically genetic research on this subtype of gastric cancer. Although some of the reported diffuse types of gastric cancers may be SRCC[3,12], high content of other tumor cells may greatly impact on the molecular characterization of signet-ring cells. Indeed, specific genetic profiles

**Fig. 3** Somatic copy number variations and structure variation in HSRCC. **a** Somatic structure variations of all patients were combined and illustrated with CIRCOS plot. Translocations between *CLDN18* and *ARHGAP26/6* were highlighted in red line. Recurrent mutated genes (SNV/INDEL only) were indicated in the outlier of rim and the SMGs were labeled in red (including *ARHGAPs* and *ARHGEFs*). Cytoband was illustrated in the inner ring, followed by illustration of copy number alteration (orange represent gain and green represent loss). Structure variations were shown inside of the CIRCOS plot, red lines represent the recurrent *CLDN18-ARHGAP26/6* fusions, green and black lines represent inter-chromosomal and intra-chromosomal translocations. **b** Illustration of breakpoint of *CLDN18* and *ARHGAP26/6* in DNA level (upper arrows) and RNA level (lower arrows). Fusions of *CLDN18* with exon 10 of *ARHGAP26*, exon 12, and *ARHGAP6* were indicated with green, purple and orange upper arrows in DNA, respectively. The junctions of *CLDN18* and *ARHGAPs* in RNA level were indicated with red, yellow, and blue lower arrows or dashed lines in the gene map demonstration and Sanger sequencing graphs, respectively

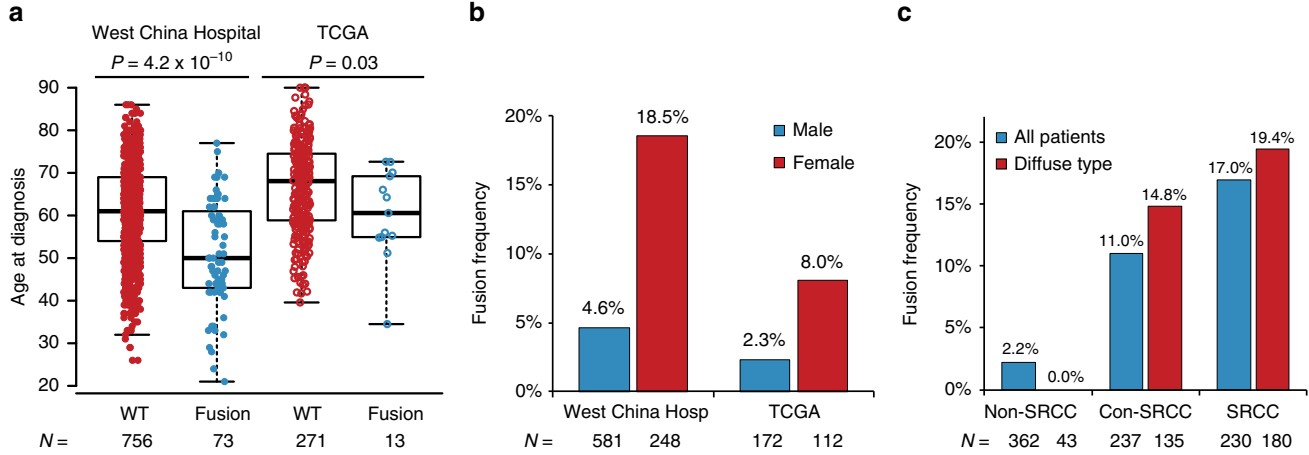

**Fig. 4** Clinical characteristics of patient with *CLDN18-ARHGAP26/6*. 829 Patients (combining 32 patients for whole-genome sequencing and 797 patients for validations) with distinct clinical features in terms of age, gender, and signet-ring cell content were illustrated in **a–c**, respectively. *P* value was estimated by using logistic regression, and the center values represent median age; WT, wildtype. Each box includes data between the 25th and 75th percentiles, with the horizontal line indicating the median. Whiskers indicate the maximal and minimal observations that are within 1.5 times the length of the box

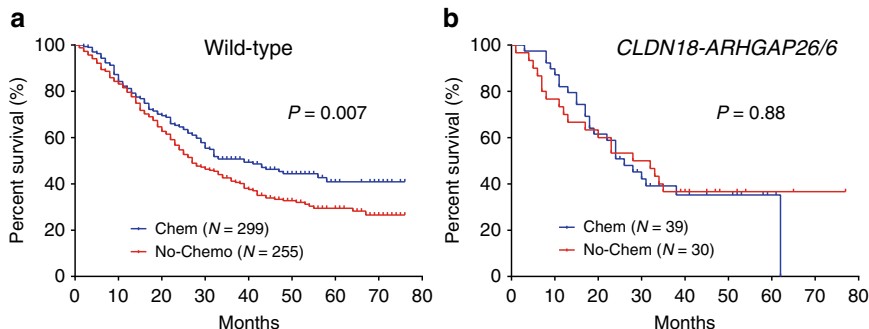

**Fig. 5** Impact of *CLDN18-ARHGAP* fusion on chemotherapy treatment outcomes. Survival curves with/without chemotherapy treatment in patients without *CLDN18-ARHGAP* fusion (**a**) or fusion burden (**b**) at stage III and IV. *P* value was estimated by using Cox model

including mutation rate, SCNA profile, altered gene cluster, etc.[3,7], have been identified in different subtypes of gastric cancer. To our knowledge, this is the first whole genomic screening on high-content signet-ring cell tumors. HSRCC belongs to diffuse type of gastric cancer, and consistently has low mutation rate, high frequency of *TP53* alterations[3,7,11,12], foci deletion in *FHIT*, amplifications of multiple oncogenes (e.g., *FGFR2*, *CD44*, and *CCNE1*), and enriched mutations in cell adhesion-related genes[12]. However, high frequency of amplification in *MYC* and *BCAS1*, and low mutation rate in *ARID1A* and *RHOA* are noticed in our study, suggesting genetic differences between HSRCC and other subtypes of diffuse gastric cancer. High frequency of gastric cancer specific fusions (i.e., *CLDN18-ARHGAP26/6*) has been detected in our HSRCC, which has also been reported in TCGA research[3]. Besides the reported fusion pattern of *CLDN18-ARHGAP26* in TCGA, a rare un-reported case of *CLDN18*/exon4-*ARHGAP26*/exon11 has been identified in our patient cohort, although the last exon (i.e., exon5) is spliced out, whole region of the most conserved domain (i.e., Claudin superfamily) of CLDN18 has been retained, indicating the similar role of such fusion pattern as others. Given that these fusions retained the Rho-GAP domain of ARHGAP26/6, we considered that CLDN18-ARHGAP26/6 may drive the endocytic membrane turnover at tight junction by transient interaction with the Rho-GAP domain of ARHGAP26/6. Because CLDN18 locates at cell membranes through its four transmembrane domains, and is highly expressed in gastric mucosa epithelium cells, CLDN18-ARHGAP26/6 fusion protein would result in over-presentation of Rho-GAP domain close to cell surface.

Patients with *ARHGAP26/6* fusion have distinct clinical characteristics, and *ARHGAP26/6* fusion is enriched in patients with SRCC subtype, as well as younger age, higher female/male ratio, advanced tumor stage, which is in parallel with that for SRCC patients, thus can partially account for their clinical relevance. In our patient cohort, *ARHGAP26/6* fusion can be detected in 17% of all SRCC patients, or up to 35% of female SRCC patients with no more than 45 years old at diagnosis. Although it cannot represent all the SRCC, the frequency of such fusion is much higher than most of the reported driver mutated and druggable genes in tumorigenesis in other types/subtypes of cancers (e.g., ALK fusion), indicating its important role on tumorigenesis of SRCC. More importantly, patients with *CLDN18-ARHGAP26/6* fusion are resistant to current chemotherapy strategy, which not only partially explains the worse treatment outcome of SRCC, but also suggests detection of such fusion is important to determinate therapy strategy usage in precision medicine era. Actually, recent functional experiments exhibit the cellular characteristic changes in *CLDN18-ARHGAP26* introduced cells, including reduced cell–EMT adhesion/loss of epithelial integrity, and cell proliferation[16]. We found a significant increasing trend of cell

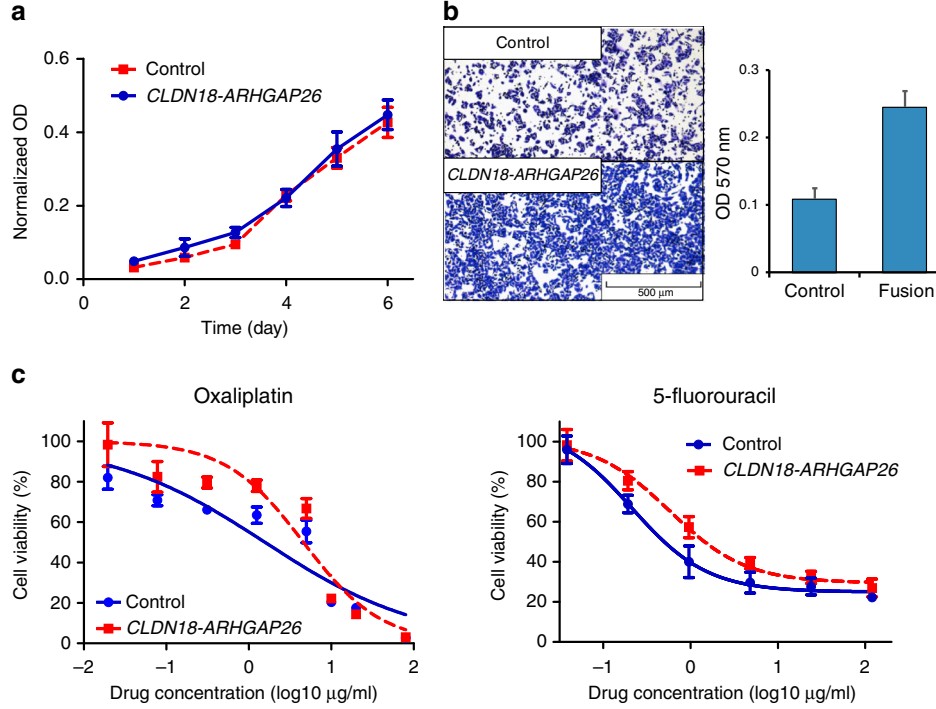

**Fig. 6** Impact of *CLDN18-ARHGAP26* overexpression in cell line. Cell proliferation (**a**), cell migration (**b**), and drug response to oxaliplatin/5-fluorouracil (**c**) was estimated in *CLDN18-ARHGAP26* stably expressed BGC-823 cells and its matched control. Each experiment has been replicated for three times, and data is presented as mean ± SD

**Table 2 Impact of *CLDN18-ARHGAP26/6* fusion on chemotherapy treatment outcomes**

| Patients[a] (No. of patients with/without chemotherapy) | Univariate | | Multivariate[b] | |
|---|---|---|---|---|
| | HR (95% CI) | *P* value | HR (95% CI) | *P* value |
| All (*N* = 422/382) | 1.26 (1.03–1.53) | 0.02 | 1.37 (1.13–1.67) | 0.002 |
| Fusion (*N* = 41/31) | 0.98 (0.54–1.79) | 0.95 | 1.03 (0.55–1.94) | 0.92 |
| Wildtype (*N* = 381/351) | 1.30 (1.05–1.60) | 0.01 | 1.41 (1.15–1.75) | 0.001 |

*P* values and HRs were estimated by Cox regression model
*HR* hazard ratio, *95% CI* 95% confidence interval of the risk ratio
[a]804 out of 829 patients (combining 32 patients for whole genome sequencing and 797 patients for validations) have full follow-up information, with platinum/fluoropyrimidines treatment or no chemotherapy treatment at all
[b]TNM stage and SRCC were adjusted in multivariate analyses

migration ability in *CLDN18-ARHGAP26* overexpressed gastric cancer cells, but no significant changes in cell growth, probably due to different selections of cell lines from Yao's reports (e.g., they used breast cell line MCF10A to illustrate the reduced cell proliferation, while we used multiple gastric cancer cell lines). Importantly, we found significant resistance to oxaliplatin and 5-fluorouracil after introduction of *CLDN18-ARHGAP26 in vitro* with different infection systems (Supplementary Fig. 13a-d), providing the possible explanation of poor drug response of patients with such fusion. However, inconsistence of drug response changes in AGS cells with stably expressed *CLDN18-ARHGAP26* fusion (Supplementary Fig. 13e and 13f). We noticed that AGS cells have a verified RHOA mutation in its conserved GTPases domain according to COSMIC database (version 81, http://cancer.sanger.ac.uk/cosmic/), while RHOA were considered to be mutually exclusively mutated with *CLDN18-ARHGAP26/6* in gastric cancer patients from TCGA, indicating RHOA mutation and CLDN18-ARHGAP26/6 may function with

the similar mechanism in gastric tumorigenesis as well as drug response.

We firstly established the clinical relevant of *CLDN18-ARHGAP26/6* in gastric cancer. However, due to the lack of comprehensive clinical information of the chemotherapy treatment outcomes in the large scale of genetic studies in gastric cancer (e.g., TCGA), more independent validation studies are needed to determine the prognostic significance of both SRCC status and *CLDN18-ARHGAP*, and their impact on determination of personalized treatment strategy. Meanwhile, the biologic mechanisms of drug resistance induced by *CLDN18-ARHGAP26* should also be investigated in the future, so as to provide a way to reverse such resistance. Interestingly, ARHGAP26 is well known as a GTP-activating protein that enhances conversion of RHO GTPases to its GDP state[31], and directly binds to the downstream cell adhesion-related genes (e.g., RHOA)[32,33]. *CLDN18-ARHGAP26/6* fusion could thus be considered as target for drug screening, and it might be possible to develop novel

therapeutic strategies to treat *CLDN18-ARHGAP26/6* burdened SRCC patients preciously.

Overall, this study provides additional insights into the clinical and genomic features of SRCC, and highlights the prognostic significance of frequent *CLDN18-ARHGAP26/6* fusions.

## Methods

**Patients and specimens**. All the consecutive primary gastric cancer patients in this study underwent surgical treatment of gastrectomy from 2006/01 to Dec 2012/12, in the Department of Gastrointestinal Surgery, West China Hospital, Sichuan University. The clinical information was retrospectively collected from the prospective database of gastric cancer (Supplementary Table 1), and the tissue samples were collected from the biorepository, National Key Laboratory of Biotherapy, West China Hospital. Patients' selection procedures were listed in the Supplementary Fig. 1. Patients were classified into three groups in terms of signet-ring cell content: non-SRCC (no signet-ring cell at all), con-SRCC (containing <50% of signet-ring cells in pathologic specimen), and SRCC (containing >50% of signet-ring cells) (Supplementary Fig. 5). Patients from 2012/01 to 2012/12 undergoing whole-genome sequencing screening met the following criteria: (1) with >80% presence of signet-ring cells; (2) both the tumor and matched control samples were available (Supplementary Table 5 and Supplementary Fig. 5d. 797 additional tumor samples were collected from 2009/1–2014/12 as validations (Supplementary Data 6).

This study was approved by Ethics Committee of West China Hospital, Sichuan University (2014, no. 215), and informed consent was obtained from patients or their guardians, as appropriate.

**Treatment strategy and clinical information**. The surgical treatments were performed according to the treatment guidelines published by the Japanese Gastric Cancer Association[34]. The postoperative chemotherapy was recommended for patients with advanced tumor stages. Combinations of fluoropyrimidine and platinum regimens were used as first-line postoperative chemotherapy treatment strategies.

The following clinical information were retrieved: gender, age, tumor size, tumor location, tumor grade, residual degree[34], T stage, N stage, M stage, and TNM stage[35]. The tumor grade and tumor stage were diagnosed according to the Union for International Cancer Control/American Joint Committee on Cancer[35]. The tumor subtypes were classified according to Lauren's classification, which was started to be routinely characterized after 2012/01[5]. Other pathological characteristics, such as nervous invasion, capillary invasion and extranodal metastasis, were evaluated according to the Japanese Gastric Cancer Classification[36].

Postoperative outpatient follow-up was done routinely (every 3 months during the first two years and then every 6 months for the last 3 years). Follow-up information was updated on January 1, 2016. Finally 1703 out of 1868 (91.2%) patients had fully postoperative follow-up information, with 44 months median follow-up duration (we calculated the follow-up duration from the date of surgery to 1/1/2016 if the patients were still alive, otherwise to the date of deaths).

**Statistics of clinical characteristic and genetic alteration**. Statistical analysis was performed by SPSS statistics software, version 20.0 (SPSS, Chicago, IL, USA) or R (Version 3.2.2). The continuous variables were tested for normal distribution before analyzing by one-way ANOVA test. The ranked variables were assessed by the Log-rank test or Kruskal–Wallis test. The categorical variables were taken with the Pearson's Chi-square test (or Fisher's exact test). Impact of clinical characteristics and genetic alterations on survival outcomes were estimated by using Kaplan–Meier method, Cox proportional hazard modeling. Associations of genetic alterations with clinical characteristics were estimated by logistic regression. A two-tailed *P* value of less than 0.05 was regarded as statistically significant.

**Nucleic acid preparations and whole genome sequencing**. DNA and RNA extraction was performed from the same tissues by using the AllPrep DNA/RNA Mini Kit (Qiagen), and evaluated by using 2100 Bioanalyzer (Aglient Technologies). DNA samples from 32 pairs of HSRCC patients passed quality control for WGS. The standard protocols from Illumina were followed to construct sequencing library for WGS, and submitted to Illumina Hiseq X10 platform to generate sequencing data with 2 × 150bp reads in Fastq format. The detailed sequencing information for each sample has been listed in Supplementary Table 5, and all the following bioinformatics tools were used in the default setting. All the cleaned reads were aligned to human genome reference (GRCh37) using BWA software (version 0.7.10)[20]. Picards tools (http://broadinstitute.github.io/picard version 2.0.1) were used to remove those PCR duplicates in BAM files. Genome Analysis toolkit (GATK, version 3.5)[37] was employed to call variants and small insertion/deletions (INDELs) from BAM files. Averagely 4,247,438 SNVs and 979,980 INDELs were identified in each sample, among which 83.7% SNVs and 51.4% INDELs were annotated as polymorphisms in dbSNP137. Meanwhile, we

predicted structure variations (SVs) with CREST (version 1.0)[38] and copy number variations (CNVs) with CONSERTING[39] using default parameters.

**Somatic alterations identification**. Somatic SNVs and INDELs were identified by comparing tumors and their matched controls, using MuTect software[40] and Varscan2 (version 2.40)[41], respectively. 8658 somatic SNVs 4325 INDELs were identified per patient, in which 72.5 SNVs and 11.2 INDELs were located in coding or splice regions on average, respectively. Several false-positive SNVs and INDELs were removed through manual inspection by using visualization tool IGV (version 2.3.67)[42]. MSI was determined by WGS-based approach according to previous reports.[20] With the matched normal sample as references, somatic SVs and SCNAs were estimated and illustrated with CIRCOS (version 0.69)[43], and IGV, respectively. Additionally, significantly mutated genes were identified by using MuSic2 with the standard procedures[44].

**Validation of the common gene fusion**. RNA from the 32 pairs of HSRCC tumor/control samples were subjected to reverse-transcribed PCR validation (F: TGGTGCGGCTCTGTTCGT/R: TGGGTCTTTATCTCCCATTCA and F: TGGTGCGGCTCTGTTCGT/R: TCGTCCCTCTGCAAGTCC for *CLDN18-ARHGAP26* and *CLDN18-ARHGAP6* fusion, respectively). Additionally, RNA was extracted from expanding 835 tumor samples of gastric cancer patients, 797 of which were in good condition (260/280 > 2, and RIN value > 5) (Supplementary Data 6), including 362 non-SRCC, 237 con-SRCC, and 198 SRCC, same RT-PCR condition was proceeded to identify *CLDN18-ARHGAP6/26* fusions.

**Pathway enrichment and PPI analyses**. 949 mutated genes were submitted to DAVID (version 6.7) for gene enrichment analyses in non-hypomutant and hypomutant group, respectively[45]. Further adhesion genes-related PPI network was built by using data set from PrePPI[46], and modified based on the interactions between the mutated cell adhesion and rest genes.

**Public information downloading**. Information of gastric cancer patients from TCGA were downloaded for validation from https://tcga-data.nci.nih.gov/tcga/, including genetic alterations and clinical characteristics.

**Functional experiments with fusion introduced cells**. The construct of pMXs-Puro-CLDN18-ARHGAP26 was purchased from Addgene (http://www.addgene.org/69465/). The fragment of *CLDN18-ARHGAP26* fusion was got by using BamHI and NotI sites, and cloned into pCDH-CMV-MCS-EF1-copGFP, and confirmed by Sanger sequencing. Four available gastric cancer cell lines were picked for checking *CLDN18-ARHGAP26* fusion and CHD1 mutations. MKN-45 was excluded because it had *CDH1* mutation. The rest three cell lines (i.e., BGC-823, AGS, and MKN-74) were infected with either control (empty vector) or *CLDN18-ARHGAP26* fusion contained retrovirus, which was packaged with helper vectors (PCMV-VSV-G and PCL-Eco), and sorted with GFP signal through flow cytometer. For validation of the drug resistance, we also used retrovirus system to infect BGC-823 by using MSCV-expression plasmid, and selected the positive pools with puromycin. MKN-74 cell line was purchased from Japanese Collection of Research Bioresources Cell Bank, BGC-823 was purchased from Cellbank of Shanghai Institutes for Biological Sciences, AGS cell line was purchased from ATCC. All cells lines were treated with Plasmocin (Invivogen) for 2 weeks before storage in liquid nitrogen, and routinely tested for mycoplasma with Mycoplasma Stain Assay Kit (Beyotime).

Transwell assay was used to evaluate cell migration ability. Briefly, 1 × 10^5 cells in 200 μl quench medium (5% BSA 1640 mediun) were seeded in 12-mm Transwell inserts (Corning), and plated into a 24-well plate with 800 μl 10% FBS 1640 medium. After 48 h incubation, The insertions were fixed with 4% paraformaldehyde for 30 min, stained with 1% crystal violet for 30 min, and followed by microscope examination after three times of washing. 33% acetic acid was added to the 24 wells for eluting the crystal violet of the transwell inserts, then the absorbance of the eluent was measured at 570 nm by microplate reader to indirectly reflect cell numbers. All the experiments have been replicated for three times.

MTT assay was used to estimate drug response. Briefly, 3000 cells were seeded each well of 96-well plates with 100 μl of 10% FBS 1640 medium, and treated with gradient concentrations of oxaliplatin or 5-fluorouracil (Sigma) on the second day. After 2 (for AGS and BGC-823) or 3 (MKN-74) additional days of incubation, 10 μl MTT (5 mg ml$^{-1}$) was added into each well and incubated for 3 h. Afterwards, medium was removed and dimethyl sulfoxide (100 μl) were added. The absorbance was measured at 570 nm with microplate reader.

**Data availability**. The authors declare that all data supporting the findings of this study are available within the article and its Supplementary Information files or from the corresponding author upon reasonable request. The sequence reported in this paper has been deposited in European Genome-Phenome Archive (EGA) database (accession no. EGAS00001002668), Genome Sequence Archive (http://bigd.big.ac.cn/gsa, accession no. PRJCA000666), and GCBI database in China (https://www.gcbi.com.cn/dataplus/html/index, accession no. GCD1000).

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

## Acknowledgements

This work was supported by (1) National Key Research and Development Program of China (No. 2016YFC0905000 [2016YFC0905002], No. 2016YFC0906000 and No. 2017YFC0909300); (2) National Natural Science Foundation of China (No. 81522028, No. 81673452, No. 81400120, and No. 81402561); (3) Sichuan Province Youth Science and Technology Innovative Research Team (No. 2015TD0009); (4) 1.3.5 project for disciplines of excellence, West China Hospital, Sichuan University; (5) Young Scientist start fund of Sichuan University (No. 2015SCU04A44). Heng Xu, Lunzhi Dai, Wei Cheng, and Dan Xie are supported by the grant from "The Recruitment Program of Global Young Experts" (known as "the Thousand Young Talents Plan")

## Author contributions

H.X., J.K.H., Y.Q.W., Y.P. designed the study; W.H.Z., D.J., X.H.S., L.Y.Z., X.Z.C., X.L.C., Y.Z., J.K.Z., Y.F.A., K.Y., J.P.L., Y.L.W., M.Q. collected the clinical information; Y.S., W. H.Z., S.Y.Z., S.L.Y., X.Y.X., D.Y.Z., X.Y.Z., L.O., L.L.F., L.Z., G.H., H.S.Y., P.Q.W., H.C., P.W., B.D., B.L., L.Y., W.C., D.X., H.X., X.H.F., L.Z.D., Y.L.Z. analyzed and interpreted the data; Q.Q.H., J.L.Z., D.D.Y., F.L. conducted the cell experiments; Y.S., W.H.Z., S.Y.Z., W.Z., Y.S.W., H.X. conducted the statistical analysis; J.K.H., W.H.Z., B.Z., H.C., H.Y.S. provided the technical and material support; J.K.H., H.X., W.M.L., Y.Q.W., Z.G.Z.

supervised this study; All authors contributed to the writing of the manuscript and final approval.

## Additional information

**Competing interests:** Hongye Sun, Hua Cheng, and Bin Zheng are employees of WuxiNextCODE. The remaining authors declare no competing interests.

