## [Peer Review File · Nature Communications]

Reviewers' comments:

Reviewer #1 (Remarks to the Author):

This paper provides a survival and drug response analysis on a very large cohort of gastric cancers, with an emphasis on signet ring cell carcinomas (SRCC). A series of 32 'pure' SRCCs were then subjected to WGS and a common CLDN18-ARHGAP26/6 fusion followed up in a further 797 patients. The paper provides very useful insights into SRCC and, in particular, the significance of the CLDN18-ARHGAP fusions. These fusions have been reported before, but this work clarifies their significance and frequency. The apparent chemo-resistance of SRCCs and the fusions will provide very useful insights for the future interpretation of gastric cancer clinical trial results.

- Supplementary Figure 5 shows the survival curves for IGC and DGC (out to about 40 months). Looking at Fig1D, there is some evidence for a survival difference for the SRCC patients at about 40 months which is lost later. It is possible that the observed DGC survival advantage may not have lasted to >60 months. The difference in follow up times should be clearly noted in the results.
- The HSRCC abbreviation is not defined in the results sections.
- Line 249 and 257: 'and etc' should be completed.
- Line 309 what are the consequences of the cryptic splice site activation?
- Line 417: 5FU?
- The modelling of the CLDN18-ARHGAP-CDC42 complex is speculative and should not be introduced for the first time in the discussion. If it is to be discussed, further justification would be valuable. Other effects of the fusion are possible. What is the red compound in Suppl Fig. 13? What is its significance?

Overall, the manuscript provides a very valuable addition to the body of knowledge on gastric cancer.

Parry Guilford

Reviewer #2 (Remarks to the Author):

The authors evaluated the significance of signet ring cell gastric cancer and conducted WGS and evaluated the clinical implication of CLDN18-ARHGAP26/6 fusion which was identified in TCGA.

1. The result from clinical data of 1868 patients with gastric cancer was retrospective design, the classification of con-SRCC (<50%) and SRCC (>50%) was decided arbitrarily, and the result was not validated in other cohort. Who and how many pathologists decided the con-SRCC and SRCC? If more than two, how was the concordance of them? What was the reason the authors decided the cutoff point of 50%?

2. It is well known that diffuse type of gastric cancer is related that poor prognosis and chemotherapy resistance, even though there has been lots of definition of diffuse type of gastric cancer such as Molecular-diffuse type (Patric Tan et al. Gastroenterology), MSS-EMT (Cristescu et al., Nature Med.), Genomic Stable (TCGA). In this sense, there is no novel finding in this study.

3. Although the authors identified the clinical significance of CLDN18-ARHGAP26/6 fusion in clinical data, and they validated it in in vitro model without any biologic evaluation for the reason of it.

4. Although % % of SRC was related to CLDN18-ARHGAP26/6 fusion, only 17% of SRCC (>50% of SRC) has this fusion. Thus this fusion is difficult to represent the biology of SRC gastric cancer.

Minor comments

Figure 2, frequency (%) of CLDN18-ARHGAP26/6 seemed to be wrong. (it seemed around 20% rather than 25%)

797 patients ? (in abstract) or 829 patients (main text) for fusion gene validation?

It was difficult to understand what sup. Table 10 meant. Title for the table and content seemed to be discordant.

The enrolled period for clinical information was 7 years (2006-2012), and the last f/u day was 2016 and 90% of patients were completed followed up. However median f/u was only 44 months.

In usual, the prognosis of patients who received chemotherapy is worse than that of not received chemotherapy in univariable analysis. It is because chemotherapy tends to be applied for patients with more advanced stage. In this sense, the result seemed to be out of expectation. How do you think about it? And how was the relationship between TNM stage and chemotherapy or not?

Reviewer #3 (Remarks to the Author):

Comments

The work by Shu and colleagues describes a whole genome sequencing (WGS) analysis of a histologically special subtype of gastric cancer, signet-ring cell carcinoma (SRCC) that has a bad prognosis. The authors screened the clinical characteristics of 1,868 Chinese gastric cancer patients and found associations between SRCC and clinical features that largely have been described before. The authors performed WGS on 32 SRCC cases with high content of signet-ring cells and identified in eight tumors (25%) the fusion gene CLDN18-ARHGAP26/6. This fusion has been described earlier to be recurrent in gastric cancer and its enrichment in genomic stable and thereby diffuse gastric cancer, where SRCC is a sub category, has been noted. The authors screened by RT-PCR a collection of 797 gastric cancers and identified an additional 65 CLDN18-ARHGAP fusions. This large number of fusion positive cases allowed interesting clinical correlation analyses with significant correlations with signet-ring cell content, age at diagnosis, gender, TNM stage and survival. The authors found no benefit of the fusion positive patient group from chemotherapy which is supported by experiments where the fusion gene was introduced in gastric cancer cell lines.

Overall, the study is of value. The novel and most interesting findings are the identification of the CLDN18-ARHGAP fusions being frequent in SRCC, the correlations between the fusions and clinical characteristics as well as the resistance of this patient group to chemotherapy. However, I have the following concerns.

Major points:

1. An analysis that integrates SCNA, fusion gene, SV break point-based alterations and SNV/indel should be performed so that convergence of different alteration types on the gene and pathway level can be identified. The recurrent copy number events that affect driver genes should be incorporated into Fig. 2.
2. The functional data on CLDN18-ARHGAP26 are not consistent across cell lines and I am wondering how robust they are. Have the authors tested the expression of the fusion gene at the end of their (or parallel) experiments? The expression might go down over time. To judge the robustness of these results, this will be important. Have different clones/transfections of each cell line been used as independent experiments? This will be necessary in particular for situations where the data is inconsistent across cell lines.
3. Among the functional experiments, the results on proliferation are most consistent. However, they are in disagreement with the report by Yao et al. (2015 Cell Rep 12:272-85) where a reduced proliferation for cell lines with the fusion gene has been described. The authors should discuss this.
4. The cells of the migration assay should be quantified.
5. The drug treatment experiments for MKN-74 should be performed up to higher drug concentrations for both, oxaliplatin and 5-fluorouracil, since the slope of the viability is still going down and the remaining viability is still high. Why do the ranges of drug concentrations differ between the different

cell lines for the same drug? For oxaliplatin in BGC-823, the concentrations range from 10^{-2} to 10^2 ug/ml, while for AGS, the concentrations range from a bit more than 10^{-1} to a bit less than 10^2 ug/ml, and for MKN-74, the concentrations range from less than 10^{-1} to a bit less than 10^2 . Omitting the lowest concentration data point for BGC-823, which is lower than for the other cell lines, would make the fusion and control curves more similar. The IC50 differences between fusion and control in some cases seem to be influenced by outlier data points, e.g. MKN-74 oxaliplatin fusion at $10^{1.2}$ ug/ml. It would be more convincing if the cell viabilities of consecutive drug concentrations would fluctuate less in for BGC-823 5-FU and MKN-74 Oxaliplatin. It would be good to repeat these experiments.

6. I don't understand the purpose of modeling the structure of CLDN18-ARHGAP26 with its interaction with CDC42 in the context of this manuscript. The interaction with CDC42 is not mentioned before or after the respective sentences in the discussion on page 21. It is not clear to me why the interaction with CDC42 but not RHOA or PTK2 or PXN is modelled. I suggest to delete this part including Supplementary Fig. 13 from the manuscript or provide a rationale why this is important in the context of the other presented findings.

7. The sequencing data has to be deposited in a database with controlled access.

Minor points:

1. All acronyms, e.g. EBV, MSI, GS, and CIN, should be spelled out when they are first mentioned.
2. Supp. Fig. 1: In the patient flow chart: in the middle there are 1,467 patients excluded "due to without complete tumor issue and peri-tumor issue" (typo: tissue) and further down another 45 tumors are excluded "without both tumor tissue and peri-tumor tissue; the proportion of signet ring cells <80%". The fact that at two points samples are excluded that have not tumor and peri-tumor tissue sounds redundant and should be explained or corrected.
3. Results, 'Somatic alterations identification' section, line 188, "on average" or "a median of" should be added to the 8,656 somatic SNVs and 4,325 INDELS per patient.
4. The large Supplementary Tables 3, 6, 9, and probably also 2 should be provided as Excel spreadsheets rather than pdf files. The column "position" in Supp Table 6 is currently not displayed correctly.
5. In general, the information in the method section is a bit sparse. If bioinformatics tools were not used with default settings, this should be indicated. A table with sequencing depth, library insert sizes, number of reads, redundancy etc. should be provided as supplementary information to give an impression regarding the sensitivity for the identification of structural variations. For the functional work, it is not clear whether empty vector, a vector with a mock insert or no vector has been used as control. How many replicates have been used for the functional experiments?
6. More information in the figure legends would be helpful, i.e. number of all and stage III/IV patients in Fig. 1; the circos plot in Fig. 3 represents the summary of all 32 samples, this should be made clear; the recurrent mutated genes in the outer rim of Fig. 3 are only based on SNV/indels? Or does it also include recurrent SCNAs? What does the red color for some gene names in the outer rim mean? The ring inside the cytoband is copy number, I assume. Please clarify. Indicate patient numbers for Fig. 4B. Why is the sum of the patients in Fig. 4C not adding up to 756 + 73 as in Fig. 4A? It should be made clear which patients have been used for the individual panels of Fig. 4. Please provide patient numbers in the legend for or within Fig. 5. On how many experiments are the panels based on? What do the error bars represent (SD, SEM)?
7. For Fig. 1 C and D, different colors should be selected that allow a better discrimination of the 4 groups.
8. In the title of Fig. 2 the information should be included that it is the landscape of HSRCC (not general gastric cancer).
9. The headers of a number of supplementary tables are not very convenient to read due to many line breaks. A vertical or tilted orientation of the headers might help.
10. In Results section "Genomic alterations of SRCC identified by WGS" lines 277 and 278 has some

error. I assume that "... with all available 32 ... characteristics were observed (te increased signi..." should be deleted.

11. What are the other three significantly mutated genes besides TP53, CDH1, and PIK3CA?

12. In Results section "Genomic alterations of SRCC identified by WGS" in lines 290/291, the individual mutated genes should be listed that belong to the GTPase-activating protein (n=4) and GDP/GTP-exchange factor (n=5) categories.

13. The panel labeling in Supp Fig. 10 is missing. At the bottom of page 16 the reference to Supp Fig. 10 should be 10B-F, not 10B-8F.

14. The authors identified a case where CLDN18 exon 4 is fused to ARHGAP26 exon 11. This has not been described before. Can the authors discuss whether the absence of exon 5 results in a loss of any functional domain of CLDN18?

15. A native speaker should go through the manuscript.

Reviewer #1 (Remarks to the Author):

This paper provides a survival and drug response analysis on a very large cohort of gastric cancers, with an emphasis on signet ring cell carcinomas (SRCC). A series of 32 'pure' SRCCs were then subjected to WGS and a common CLDN18-ARHGAP26/6 fusion followed up in a further 797 patients. The paper provides very useful insights into SRCC and, in particular, the significance of the CLDN18-ARHGAP fusions. These fusions have been reported before, but this work clarifies their significance and frequency. The apparent chemo-resistance of SRCCs and the fusions will provide very useful insights for the future interpretation of gastric cancer clinical trial results.

- Supplementary Figure 5 shows the survival curves for IGC and DGC (out to about 40 months). Looking at Fig1D, there is some evidence for a survival difference for the SRCC patients at about 40 months which is lost later. It is possible that the observed DGC survival advantage may not have lasted to >60 months. The difference in follow up times should be clearly noted in the results.

Thanks for the reviewer's comment and sorry for not explained clearly in our manuscript. Since the Lauren's classification is routinely evaluated only after 2012, which can be checked in Supplementary Table 1 (84.4% patients enrolled in this study don't have Lauren's classification). Therefore, for supplementary Figure 5, we only illustrated the patients enrolled after 2012. We have added notes in the legend of this figure as well as Methods part (page 20).

- The HSRCC abbreviation is not defined in the results sections.

Thanks for the reviewer's notification, we have added the definition in for HSRCC in results sections (page 9)

- Line 249 and 257: 'and etc' should be completed.

Thanks for the reviewer's notification, we have added all the significant clinical features (page 8)

- Line 309 what are the consequences of the cryptic splice site activation?

Thanks for the reviewer's suggestion, we have added the description (page 11)

- Line 417: 5FU?

Sorry for the mistakes, we have added the word "5-fluorouracil" (page 17)

- The modelling of the CLDN18-ARHGAP-CDC42 complex is speculative and should not be introduced for the first time in the discussion. If it is to be discussed, further justification would be valuable. Other effects of the fusion are possible. What is the red compound in Suppl Fig. 13? What is its significance?

Thanks for this suggestion which is also mentioned by another reviewer. We have deleted the Supplementary Figure 13 and the according description as requested.

Overall, the manuscript provides a very valuable addition to the body of knowledge on gastric cancer.

Reviewer #2 (Remarks to the Author):

The authors evaluated the significance of signet ring cell gastric cancer and conducted WGS and evaluated the clinical implication of CLDN18-ARHGAP26/6 fusion which was identified in TCGA.

1. The result from clinical data of 1868 patients with gastric cancer was retrospective design, the classification of con-SRCC (<50%) and SRCC (>50%) was decided arbitrarily, and the result was not validated in other cohort. Who and how many pathologists decided the con-SRCC and SRCC? If more than two, how was the concordance of them? What was the reason the authors decided the cutoff point of 50%?

We apologized for not mentioning the derivation of SRCC concept. Actually, definition of SRCC with signet ring cell contents > 50% has been described according to World Health Organization (WHO) classification (Hamilton SR, *et al*, WHO classification of tumours of the digestive organs. IARC; Lyon, pp: 134–146; Watanabe H *et al*: Histological Typing of Oesophageal and Gastric Tumours: WHO International Histological Classification of Tumors (ed 2). Berlin, Germany, Springer-Verlag, 1990), and widely adopted (e.g., Piessen G *et al*, *J Clin Oncol*. 2013; Taghavi S *et al*, *J Clin Oncol*. 2012). Besides SRCC patients, signet ring cells can be found in tumor sample from some patients but don't exceed 50%, we noticed that these patients also have a lower survival rate and sensitivity to chemotherapy than patients with no signet ring cell detected, therefore, we used con-SRCC to define these patients in this manuscript. We have described the classification of SRCC and cited the reference (page 7 and 8)

2. It is well known that diffuse type of gastric cancer is related that poor prognosis and chemotherapy resistance, even though there has been lots of definition of diffuse type of gastric cancer such as Molecular-diffuse type (Patric Tan *et al*. Gastroenterology), MSS-EMT (Cristescu *et al*., Nature Med.), Genomic Stable (TCGA). In this sense, there is no novel finding in this study.

Thanks for the reviewer's comments. We used diffused type of gastric cancer for analyses as positive controls, which is only illustrated in supplementary figures. Actually, we are focusing on the subtype of SRCC in this manuscript but not diffuse type of gastric cancer. We firstly defined con-SRCC (with signet ring cell content between 0 to 50%) in our study, and found that the survival rate and sensitivity to chemotherapy of this group of patients are mostly rank between non-SRCC and SRCC, while HSRCC (with signet ring cell content > 80%) have the worst treatment outcomes, indicating the survival rate is in parallel with the signet sing cell contents. Although not surprisingly, it is a novel finding only mentioned in our current study. Additionally, although a different classification criteria is used for SRCC definition, all of SRCC patients belong to diffused type of gastric cancer. However, diffuse type of patients can get significant benefits from chemotherapy treatment (supplementary figure 5), which is not true for SRCC patients based on previous and our results. SRCC was therefore considered as an independent prognostic factor even in patients with diffuse type (page 9). In addition, we also checked the frequency of CLDN18-ARHGAP26/6 fusion within the diffused type, and noticed a consistent enrichment of such fusion in SRCC (information has been provided in the modified figure 4C and described in

page 12). Moreover, Since SRCC status are reported to be associated with gender, age and stage, we conducted multivariate analyses and found that the association of *CLDN18-ARHGAP* fusion with these clinical characteristics can only partially explained by SRCC status (Supplementary Table 11). For instance, the frequency of *CLDN18-ARHGAP* fusion raised to 12.1% and 24.4% in male and female patients with SRCC, respectively (page 13). These novel findings suggested that subgroups can be defined within diffuse type in terms of morphologic and molecular features and related to treatment outcomes.

3. Although the authors identified the clinical significance of *CLDN18-ARHGAP26/6* fusion in clinical data, and they validated it in in vitro model without any biologic evaluation for the reason of it.

Thanks for the reviewer's comments. In previous reports (Yao et al. 2015 Cell Rep), the possible pathway of the *CLDN18-ARHGAP* on migration has been mentioned, which we have discussed in our manuscript. We totally agree with the reviewer that we only identified the clinical relevant of *CLDN18-ARHGAP26/6* for drug resistance with validation of only in vitro model. We consider it very important to evaluate the possible mechanism in the future, so as to find a way to reverse such resistance. We added some discussion in page 18.

4. Although % % of SRC was related to *CLDN18-ARHGAP26/6* fusion, only 17% of SRCC (>50% of SRC) has this fusion. Thus this fusion is difficult to represent the biology of SRC gastric cancer.

Thanks for this comment. We totally agree with the reviewer's opinion that the fusion alone can't represent the biology of all SRCC. However, there're only a few examples that a single alteration can explain most of the patients with a specific cancer type/subtype. Due to the development of sequencing technology, more and more gene alterations have been found. Significantly mutated genes are an important indicator to distinguish driver with passenger altered genes. For TCGA analyses, 127 genes were considered as the driver genes for tumorigenesis (Kandoth C *et al*, Nature. 2013), except for a few genes like TP53, the highest mutation rates of most genes in a specific cancer type are lower than 10%, such as NRAS. Additionally, ALK fusion accounts for less than 5% of the lung cancer, but several target drugs have been developed and routinely used in clinical treatment. In this case, although *CLDN18-ARHGAP* fusion can only partially explain the drug resistance of SRCC, at least we take the first step to reveal the molecular features of SRCC. Additionally, if we focused on patient subgroup, ~35% of female SRCC patients with no more than 45 years old at diagnosis have this fusion, indicating that we can investigate the biology of SRC gastric cancer in this subgroup of patients. More importantly, such fusion could be considered as a novel druggable target especially in SRCC because it is considered to be involved in RHOA pathway and present in 17% of the patients. We have modified our discussion accordingly (page 17).

Minor comments

Figure 2, frequency (%) of *CLDN18-ARHGAP26/6* seemed to be wrong. (it seemed around 20% rather than 25%)

Thanks for the review, we apologize that we made a mistake here because we forget take the only one *CLDN18-ARHGAP6* into account. The frequency has been corrected.

797 patients ? (in abstract) or 829 patients (main text) for fusion gene validation?

Sorry for the confusing, we checked our abstract and main text and found that we consistently used "797" to describe the validation cohort, only mentioned "829" in Table 2 to indicate all the patients for CLDN18-ARHGAP fusion containing both the WGS and validation cohort. We have added a description for the number of "829" in Table 2.

It was difficult to understand what sup. Table 10 meant. Title for the table and content seemed to be discordant.

Sorry for the confusing, we were testing the correlation of multiple clinical characteristics with the presence of CLDN18-ARHGAP26/6 fusions in patients. The most significant features (e.g, female/male ratio, age and SRCC status) have been illustrated in figure 4. We changed the title and added the analyses procedure and explanation in this table to make it easier to be understood.

The enrolled period for clinical information was 7 years (2006-2012), and the last f/u day was 2016 and 90% of patients were completed followed up. However median f/u was only 44 months. Sorry for the confusing, we stopped follow-up if the patients were dead, so the median f/u was only 44 months. We added the explanation in the main text (page 20)

In usual, the prognosis of patients who received chemotherapy is worse than that of not received chemotherapy in univariable analysis. It is because chemotherapy tends to be applied for patients with more advanced stage. In this sense, the result seemed to be out of expectation. How do you think about it? And how was the relationship between TNM stage and chemotherapy or not?

We totally agree with the reviewer's opinion. So in our study, we used both univariable as well as multivariate analyses with TNM stage as the covariate (e.g., Table 2). In addition, we also illustrate our survival curves of patients in all stages as well as solely in stage III/IV (e.g., Figure 1, Figure 5, and Supplementary Figure 11), or even analyzed separately in terms of 4 different stages (e.g., Supplementary Figure 3). Interestingly, the impact of CLDN-ARHGAP on chemotherapy treatment outcomes is a risk factor for chemotherapy resistance independent of TNM stages (Table 2) (page 13).

Reviewer #3 (Remarks to the Author):

Comments

The work by Shu and colleagues describes a whole genome sequencing (WGS) analysis of a histologically special subtype of gastric cancer, signet-ring cell carcinoma (SRCC) that has a bad prognosis. The authors screened the clinical characteristics of 1,868 Chinese gastric cancer patients and found associations between SRCC and clinical features that largely have been described before. The authors performed WGS on 32 SRCC cases with high content of signet-ring cells and identified in eight tumors (25%) the fusion gene CLDN18-ARHGAP26/6. This fusion has been described earlier to be recurrent in gastric cancer and its enrichment in genomic stable and thereby diffuse gastric cancer, where SRCC is a sub category, has been noted. The authors

screened by RT-PCR a collection of 797 gastric cancers and identified an additional 65 CLDN18-ARHGAP fusions. This large number of fusion positive cases allowed interesting clinical correlation analyses with significant correlations with signet-ring cell content, age at diagnosis, gender, TNM stage and survival. The authors found no benefit of the fusion positive patient group from chemotherapy which is supported by experiments where the fusion gene was introduced in gastric cancer cell lines.

Overall, the study is of value. The novel and most interesting findings are the identification of the CLDN18-ARHGAP fusions being frequent in SRCC, the correlations between the fusions and clinical characteristics as well as the resistance of this patient group to chemotherapy. However, I have the following concerns.

Major points:

1. An analysis that integrates SCNA, fusion gene, SV break point-based alterations and SNV/indel should be performed so that convergence of different alteration types on the gene and pathway level can be identified. The recurrent copy number events that affect driver genes should be incorporated into Fig. 2.

We totally agree with the review's suggestion, and made the revision accordingly for figure 2. For integration of different alteration type, it's difficult to determine the real potential functional genes involved in the large fraction of SCNAs, we thus only selected the foci alterations containing less than three genes for analyses. Cell adhesion still ranks the top of the significantly altered gene pathway category in the updated Supplementary Table 10 (page 11)

2. The functional data on CLDN18-ARHGAP26 are not consistent across cell lines and I am wondering how robust they are. Have the authors tested the expression of the fusion gene at the end of their (or parallel) experiments? The expression might go down over time. To judge the robustness of these results, this will be important. Have different clones/transfections of each cell line been used as independent experiments? This will be necessary in particular for situations where the data is inconsistent across cell lines.

We totally agree with this comments. We used retrovirus instead of lentivirus system to infect BGC-823 cells, and selected the cells with puromycin instead of GFP sorting. Around 6-folds of oxaplatin and 2-folds of 5-FU resistance were also observed. We added some comments in discussion and method part (page 17 and 23-24). We are also confused that AGS cells weren't resistant to oxaliplatin and 5-FU, we did literature and database (e.g., COSMIC) searching, and found out that AGS cells have verified RHOA mutation in its conserved GTPases domain according to COSMIC database, while RHOA were considered to be mutually exclusively mutated with CLDN18-ARHGAP26/6 in TCGA dataset, probably because both RHOA mutation and CLDN18-ARHGAP26/6 may function with the similar mechanism in gastric tumorigenesis and drug response. We added this comment into the discussion part (page 17-18).

3. Among the functional experiments, the results on proliferation are most consistent. However, they are in disagreement with the report by Yao et al. (2015 Cell Rep 12:272-85) where a reduced proliferation for cell lines with the fusion gene has been described. The authors should discuss this.

Thanks for the review's suggestions. We also noticed that the cell proliferation rate changes in

Yao's reports, however, they used different types of cancer cells, which are not gastric cancer. The fusion CLDN18-ARHGAP fusions are only identified in gastric cancer patients, we considered that gastric cancer cell lines should be better models for investigating the function of this fusion. However, no significant growth advantage can be detected in our experiments. We have added some comments in our discussion part (page 17).

4. The cells of the migration assay should be quantified.

Thanks for the reviewer's suggestions. We have quantified the migration cells accordingly, and added a new panel in Figure 6B, Supplementary Figure 12, and method description in Method part (page 24).

5. The drug treatment experiments for MKN-74 should be performed up to higher drug concentrations for both, oxaliplatin and 5-fluorouracil, since the slope of the viability is still going down and the remaining viability is still high. Why do the ranges of drug concentrations differ between the different cell lines for the same drug? For oxaliplatin in BGC-823, the concentrations range from 10^{-2} to 10^2 $\mu\text{g}/\text{ml}$, while for AGS, the concentrations range from a bit more than 10^{-1} to a bit less than 10^2 $\mu\text{g}/\text{ml}$, and for MKN-74, the concentrations range from less than 10^{-1} to a bit less than 10^2 . Omitting the lowest concentration data point for BGC-823, which is lower than for the other cell lines, would make the fusion and control curves more similar. The IC50 differences between fusion and control in some cases seem to be influenced by outlier data points, e.g. MKN-74 oxaliplatin fusion at $10^{1.2}$ $\mu\text{g}/\text{ml}$. It would be more convincing if the cell viabilities of consecutive drug concentrations would fluctuate less in for BGC-823 5-FU and MKN-74 Oxaliplatin. It would be good to repeat these experiments.

We appreciate the reviewer's suggestions. We firstly did a preliminary experiment to estimate the IC50 and drug index region with the parental cells, and determined the usage of varied drug concentration regions for different cell lines treatment. The parental MKN-74 is resistant to 5FU and Oxaliplatin, we actually used a higher drug concentration to start than the other two cell lines. However, when we used the highest drug concentration estimated according to parental cell lines, treatment of DMSO alone can significantly induce cell death. In this case, we deleted the treatment results with such drug concentration, which we considered to be not reliable. Due to the maximal dissolution limitation of the drugs, we try to use the highest drug concentration and longer treatment time for MKN-74 (72h instead of 48h), with which no obvious toxicity was observed in no treatment controls, and have redone the experiments for MKN74 cell lines (Figure 6C and Supplementary Figure 12). Also, drug resistant experiments were repeated in retrovirus system as illustrated in supplementary figure 13.

6. I don't understand the purpose of modeling the structure of CLDN18-ARHGAP26 with its interaction with CDC42 in the context of this manuscript. The interaction with CDC42 is not mentioned before or after the respective sentences in the discussion on page 21. It is not clear to me why the interaction with CDC42 but not RHOA or PTK2 or PXN is modelled. I suggest to delete this part including Supplementary Fig. 13 from the manuscript or provide a rationale why this is important in the context of the other presented findings.

Thanks for this suggestion which is also mentioned by another reviewer. We have deleted the previous Supplementary Figure 13 and the according description as requested.

7. The sequencing data has to be deposited in a database with controlled access.

Thanks for review's suggestion, we have deposited our original data on GCBI database in China (<https://www.gcbi.com.cn/dataplus/html/index>) with the series number of GCD1000, which we have added the description in page 24. In addition, we are applying for the certification from Chinese Human Resource Control Office, and will upload our data on EGA as well once we get the permission.

Minor points:

1. All acronyms, e.g. EBV, MSI, GS, and CIN, should be spelled out when they are first mentioned.

Thanks for this comments, we have added the full name of these subtypes (page 6).

2. Supp. Fig. 1: In the patient flow chart: in the middle there are 1,467 patients excluded "due to without complete tumor issue and peri-tumor issue" (typo: tissue) and further down another 45 tumors are excluded "without both tumor tissue and peri-tumor tissue; the proportion of signet ring cells <80%". The fact that at two points samples are excluded that have not tumor and peri-tumor tissue sounds redundant and should be explained or corrected.

Thanks for this notification, and we have corrected our Supplementary Figure 1

3. Results, 'Somatic alterations identification' section, line 188, "on average" or "a median of" should be added to the 8,656 somatic SNVs and 4,325 INDELS per patient.

Thanks for review's to point out this problem, we have added "on average" in page 22.

4. The large Supplementary Tables 3, 6, 9, and probably also 2 should be provided as Excel spreadsheets rather than pdf files. The column "position" in Supp Table 6 is currently not displayed correctly.

Thanks for the suggestion, and we will provide all the supplementary tables as excel format.

5. In general, the information in the method section is a bit sparse. If bioinformatics tools were not used with default settings, this should be indicated. A table with sequencing depth, library insert sizes, number of reads, redundancy etc. should be provided as supplementary information to give an impression regarding the sensitivity for the identification of structural variations. For the functional work, it is not clear whether empty vector, a vector with a mock insert or no vector has been used as control. How many replicates have been used for the functional experiments?

Thanks for the suggestion, and we have provided the sequencing information as Supplementary Table 12, and described in Methods part. All the bioinformatics tools were used with the default setting (page 21). Each experiments has been replicated for 3 times for the functional experiments (page 24).

6. More information in the figure legends would be helpful, i.e. number of all and stage III/IV patients in Fig. 1; the circos plot in Fig. 3 represents the summary of all 32 samples, this should be made clear; the recurrent mutated genes in the outer rim of Fig. 3 are only based on SNV/indels? Or does it also include recurrent SCNAs? What does the red color for some gene names in the outer rim mean? The ring inside the cytoband is copy number, I assume. Please

clarify. Indicate patient numbers for Fig. 4B. Why is the sum of the patients in Fig. 4C not adding up to 756 + 73 as in Fig. 4A? It should be made clear which patients have been used for the individual panels of Fig. 4. Please provide patient numbers in the legend for or within Fig. 5. On how many experiments are the panels based on? What do the error bars represent (SD, SEM)?

Thanks for the reviewer's suggestions, and we have added the information accordingly in our figure legends.

7. For Fig. 1 C and D, different colors should be selected that allow a better discrimination of the 4 groups.

Thanks for this suggestion, and we have changed the colors for Fig 1C and D.

8. In the title of Fig. 2 the information should be included that it is the landscape of HSRCC (not general gastric cancer).

Thanks for this suggestion, and we have modified the title of Fig. 2 (page 30)

9. The headers of a number of supplementary tables are not very convenient to read due to many line breaks. A vertical or tilted orientation of the headers might help.

We appreciated with this suggestion, and we modified the long header of multiple Supplementary Tables to make it easier to be read

10. In Results section "Genomic alterations of SRCC identified by WGS" lines 277 and 278 has some error. I assume that "... with all available 32 ... characteristics were observed (te increased signi..." should be deleted.

We are sorry for the typo, we actually meant "931 missense, 63 nonsense...." (page 9-10) We still feel confused how this part has been changed from our early version of manuscript.

11. What are the other three significantly mutated genes besides TP53, CDH1, and PIK3CA?

We have added the other three SMGs (*ERBB2*, *LCE1F*, and *OR8J1*) in the main text (page 10). Meanwhile, to avoid misleading, we deleted *KMT2C* and *KMT2D* in figure 2, which are considered to be important cancer-related genes but not SMG.

12. In Results section "Genomic alterations of SRCC identified by WGS" in lines 290/291, the individual mutated genes should be listed that belong to the GTPase-activating protein (n=4) and GDP/GTP-exchange factor (n=5) categories.

We appreciated for this comments, and added the individual gene names of ARHGAPs and ARHGEFs in the main text (page 10), and detail mutation information is provided in Supplementary Table 4

13. The panel labeling in Supp Fig. 10 is missing. At the bottom of page 16 the reference to Supp Fig. 10 should be 10B-F, not 10B-8F.

Thanks for pointing out this mistakes, and we have corrected accordingly figure and manuscript (page 11).

14. The authors identified a case where CLDN18 exon 4 is fused to ARHGAP26 exon 11. This has not been described before. Can the authors discuss whether the absence of exon 5 results in a loss of any functional domain of CLDN18?

We appreciated for this comments. Although the last exon (i.e., exon5) was spliced out for the rare new case of *CLDN18/exon4-ARHGAP26/exon11*, whole region of the most conserved domain (i.e., Claudin superfamily) of CLDN18 has been retained, indicating the similar role of such fusion pattern as others. We added this discussion in page 16

15. A native speaker should go through the manuscript.

Thanks for this suggestion, we asked a native speaker to go over all manuscript and corrected our grammar mistakes and improper expression.

Reviewers' comments:

Reviewer #1 (Remarks to the Author):

I am satisfied with the responses to my questions.

Reviewer #2 (Remarks to the Author):

One of main outcome of this study is prognosis and responsiveness of CLDN18-ARHGAP26/6 fusion, so accurate clinical information is important.

In supple table 1, the number of M1 should be same to the number of stage IV.

In S table 10, Odds ratio would be right rather than hazard ratio because the authors used logistic regression model. Also the authors need to consider multicollinearity for their multivariable analysis. They need to discuss this issue with statistician.

Supple figure 4 and 5 showed discrepant result. In S fig. 4, the benefit from chemotherapy was not observed in SRC. However, benefit was observed in diffuse type of gastric cancer rather than intestinal type. This result is out of common knowledge of gastric cancer. Also, as the authors described, most of SRC classified as diffuse type, thus those results were opposite.

Benefit from chemotherapy was observed even in Con-SRC group, and the frequency of fusion was 11.0% vs. 17.0% between con-SRC and SRCC. How about the clinical characteristics of wild type of SRC and con-SRC?

Reviewer #3 (Remarks to the Author):

Yao et al. used four different cell lines in their proliferation experiments, one of them is a gastric cancer cell line. All four cell lines showed a reduced proliferation when expressing CLDN18-ARHGAP26 compared to the original cell lines. The statement in line 337 on page 17 referring to the use of a different cell type is therefore misleading. Stating that Yao et al. used different cell lines would be appropriate.

All other points have been addressed well.

Reviewers' comments:

Reviewer #1 (Remarks to the Author):

I am satisfied with the responses to my questions.

Thanks a lot for this reviewer's valuable comments and suggestions.

Reviewer #2 (Remarks to the Author):

One of main outcome of this study is prognosis and responsiveness of CLDN18-ARHGAP26/6 fusion, so accurate clinical information is important.

In supple table 1, the number of M1 should be same to the number of stage IV.

We're so appreciated with reviewer's comments, and so sorry for the typo. The number of patients in M1 stage is 204 but not 243, which is consistent with that for patients in TNM stage. We also checked the M1 stage patient number in Table-S2, which is consistent with the sum up of different subtypes (N=90+76+38, with 204 in total).

We carefully rechecked clinical numbers in our main and supplementary tables, and no other mistake has been found.

In S table 10, Odds ratio would be right rather than hazard ratio because the authors used logistic regression model. Also the authors need to consider multicollinearity for their multivariable analysis. They need to discuss this issue with statistician.

Thanks for this suggestions, and we have discussed with statisticians. According to his introduction and suggestions about multicollinearity, we noticed that stage T, stage N, and stage M can't be adjusted company with TNM stage, and SRCC can't be adjusted company with Lauren classification, we thus redid our multivariate analyses without adjusting Lauren's classification (consider SRCC rather than Lauren's classification because limited number of patients have such information in our cohort) and TNM stage. Notes were also added in the explanation part of Table-S10. All "Hazard ratio (HR) " has been changed into "odds ratio (OR)" in Table-S10 (has changed into Table-S11 to fit the order).

Supple figure 4 and 5 showed discrepant result. In S fig. 4, the benefit from chemotherapy was not observed in SRC. However, benefit was observed in diffuse type of gastric cancer rather than intestinal type. This result is out of common knowledge of gastric cancer. Also, as the authors described, most of SRC classified as diffuse type, thus those results were opposite.

Thanks for this comments. It's absolutely right that intestinal type of gastric cancer should have a better response to diffuse type. We considered three possibilities which may explain the reason why no significant difference was observed in intestinal patients in terms of chemotherapy: ① In our research cohort (2006-2012), the number of patients with Lauren's classification is limited because it was started to be routinely characterized after 2012/01 (highlighted in page 19), thus Supplementary Figure 5 was illustrated based on only 94 (90 have follow-up information)

intestinal type and 197 (191 have follow-up information) diffuse type of patients (listed in supplementary Table 1), which may reduce the statistic power; ② treatment outcome of intestinal type is much better than diffused type, with more than 80% of survival rate before 2016. Although a trend can be observed, the follow-up duration is not long enough to reflect the differences; ③ patients with chemotherapy are tend to have a higher stage than those without, which was not considered in this analysis. Therefore, we redid the statistical calculation after adjusting stage, and got significant P value for both subtypes listed in Figure-S5. Sample sizes and method description for P value estimation was also added in the figure legends.

Another great question is that the benefit from chemotherapy was not observed in diffused type rather than SRCC. We considered this is mainly because diffuse type contains non-SRCC, con-SRCC as well, patients with these two subtypes will get benefit from chemotherapy, which further indicate the possible prognostic effect of SRCC rather than Lauren's classification for chemotherapy treatment outcomes. However, this conclusion can't be drawn after validations with large sample size in multiple independent cohorts. We added this comments in the discussion part (page 14).

Benefit from chemotherapy was observed even in Con-SRC group, and the frequency of fusion was 11.0% vs. 17.0% between con-SRC and SRCC. How about the clinical characteristics of wild type of SRC and con-SRC?

Thanks for reviewer's question, all the available clinical characteristics (including SRCC status) and CLDN18-ARHGAP26/6 fusion information is listed in supplementary Table -S5 (32 patients for WGS) and Table-S9 (797 patients for validation). Moreover, we did analysis in patients with fusion and wild-type separately with/without adjusting for TNM stage and SRCC status (Table 2), illustrating fusion-free patients can get benefit from chemotherapy regardless of SRCC status. In addition, we use "partially explains to demonstrate the impact of CLDN-ARHGAP on the worse treatment outcome of SRCC (page 16) avoid misleading.

Reviewer #3 (Remarks to the Author):

Yao et al. used four different cell lines in their proliferation experiments, one of them is a gastric cancer cell line. All four cell lines showed a reduced proliferation when expressing CLDN18-ARHGAP26 compared to the original cell lines. The statement in line 337 on page 17 referring to the use of a different cell type is therefore misleading. Stating that Yao et al. used different cell lines would be appropriate.

All other points have been addressed well.

Thanks a lot for this reviewer's valuable comments and suggestions. And we have changed our expression for the last suggestion accordingly (page 16)

REVIEWERS' COMMENTS:

Reviewer 2 and 3 were both satisfied with the revision.